# Recent Advances in Amphipathic Peptidomimetics as Antimicrobial Agents to Combat Drug Resistance

**DOI:** 10.3390/molecules29112492

**Published:** 2024-05-24

**Authors:** Ma Su, Yongxiang Su

**Affiliations:** 1College of Pharmaceutical Sciences, Soochow University, 199 Ren-Ai Road, Suzhou 215123, China; 2College of Chemistry and Environmental Engineering, Jiaozuo University, Ren-Min Road, Jiaozuo 454000, China; suyx-88@163.com

**Keywords:** antimicrobial agents, drug resistance, peptidomimetics

## Abstract

The development of antimicrobial drugs with novel structures and clear mechanisms of action that are active against drug-resistant bacteria has become an urgent need of safeguarding human health due to the rise of bacterial drug resistance. The discovery of AMPs and the development of amphipathic peptidomimetics have lay the foundation for novel antimicrobial agents to combat drug resistance due to their overall strong antimicrobial activities and unique membrane-active mechanisms. To break the limitation of AMPs, researchers have invested in great endeavors through various approaches in the past years. This review summarized the recent advances including the development of antibacterial small molecule peptidomimetics and peptide-mimic cationic oligomers/polymers, as well as mechanism-of-action studies. As this exciting interdisciplinary field is continuously expanding and growing, we hope this review will benefit researchers in the rational design of novel antimicrobial peptidomimetics in the future.

## 1. Introduction

Currently, at least 700,000 people worldwide die each year due to antibiotic resistance. The World Health Organization (WTO) predicts that without new and better treatments, this number could rise to 10 million by 2050 [1]. Drug-resistant bacterial infection weakens the efficacy of antimicrobial therapy, prolongs hospital stay of patients, and increases the probability of hospital outbreak occurrence and the death of patients [2,3,4]. The work on pathogen control and prevention is of great significance and cannot be replaced. Through genetic mutations, bacteria have evolved a set of highly complex and precise evolutionary and defense mechanisms, which are mainly divided into three categories: inherent resistance, acquired resistance, and adaptive resistance. Among them, the acquisition of drug resistance is the key way for bacteria to acquire drug-resistance genes. Bacteria can acquire drug resistance through lateral transfer of drug-resistance genes, mainly including the following: (1) self-mutation or modification; (2) intercellular resistance gene transfer; (3) decreased or changed membrane permeability; (4) hydrolase or inactivated enzyme action; (5) target protection protein; (6) active efflux system, etc. [5]. Adaptive resistance mainly includes the following: (1) metabolic changes and nutritional deficiencies; (2) changes in cell morphology; (3) biofilm, etc. [6,7]. The most common approach to combat drug-resistant bacteria is to develop new antibacterial molecules which have good selectivity and strong responsiveness that prevent the emergence of new drug-resistant bacteria; thus, it has become an urgent research area with increasing clinical demand to constantly explore and find anti-drug-resistant bacteria drugs with new structures and unique properties [8].

After a decline in the pipeline of new antimicrobial drugs in the 1980s and 1990s, daptomycin, a natural antimicrobial peptide that targets bacterial cell membranes, was approved by the FDA (U.S. Food and Drug Administration) in 2003 and immense endeavors have been made into researching antimicrobial peptides (AMPs) [9,10]. There are more than 1000 naturally occurring AMPs and their secondary structure varies [11,12]. The common feature of AMPs is that they possess both hydrophilic positively charged groups and the hydrophobic groups which make them amphiphilic [13]. While cationic groups interact with the negatively charged bacterial membranes, the hydrophobic groups play a key role in membrane penetration and destruction [14]. Therefore, AMPs should not cause widespread resistance due to their unique antimicrobial mechanisms. The assets of AMPs in clinical application include their potential for broad-spectrum activity, rapid bactericidal activity, and low propensity for resistance development, whereas possible disadvantages include their high cost, limited stability, unknown toxicology, and pharmacokinetics. Initial barriers to their success are being increasingly overcome with the development of stable, more cost-effective, and potent broad-spectrum synthetic amphipathic peptidomimetics [15]. As such, there have been extensive endeavors devoted to the development of antibiotic agents that mimic the mode of action of AMPs with enhanced potency, stability, and bioavailability, such as peptidomimetic small molecules [16], oligomers [17,18], cationic polymers [19,20], and unnatural amino acids (Figure 1) [21,22].

## 2. Small Molecule Peptidomimetics

Small molecule peptidomimetics could be a highly promising strategy since they exhibit much smaller molecular weight, easier synthesis, and better stability. Moreover, preclinical studies have been completed for several small molecules during the past decades. They are currently in clinical trials, such as **CSA-13** (**2**, Figure 1), **PMX30063** (**4**, Figure 2), and **LTX109** (**8**, Figure 3) [23,24].

### 2.1. Cholic Acid-Based Ceragenin Derivatives

As common bile acids, cholic acids (**1**, Figure 1) are inherently facially amphiphilic and are secreted into the gastrointestinal tract to aid in the solubilization of lipids [25]. Due to their amphiphilic nature, the cholic acid scaffold has been noted and expanded as ceragenins [26]. Ceragenins obtain both hydrophobic and hydrophilic surfaces on the rigid skeleton and can be prepared in large quantities which overcomes the limited source of AMPs [27]. Furthermore, ceragenins have advantages over AMPs in that they are resistant to proteolysis; hence, their stability is not affected by proteases *in vivo*. Since 1998, Savage et al. have designed, synthesized, and tested ceragenins as a novel series of antimicrobial agents [28,29,30,31]. Their research demonstrated that ceragenins exhibit broad-spectrum antibacterial activity against both Gram-negative and Gram-positive bacterial strains, especially some drug-resistant strains that were included [29]. Their results also showed that ceragenins are biocompatible in a variety of tissues and showed good *in vivo* stability. Among ceragenins, the most potent compound **CSA-13** (**2**, Figure 1) was developed as the lead compound for further preclinical and clinical studies; its ability of biofilm inhibition was evaluated as well [24,32,33,34,35,36]. In addition to excellent antimicrobial activity, drug-resistance studies demonstrated that **CSA-13** retained potent antibacterial activity against S. aureus over the course of 30 serial passages [37]. Squalamine (**3**, Figure 1) is a natural aminosterol isolated from dogfish shark with a steroidal structure similar to ceragenins. Squalamine derivatives have been developed due to their broad-spectrum activity and efficiency against multidrug-resistant bacteria [38,39]. The amphiphilicity of squalamine is composed of a sterol core with a sulfated side chain and a hydrophilic polyamine spermidine moiety bonded to a hydrophobic unit [40]. It has also been confirmed to have an antibacterial mechanism similar to AMPs, which can bind to negatively charged phospholipid molecules on the surface of the bacterial cell membrane, resulting in depolarization and rupture of the bacterial cell membrane and eventual death [41].

### 2.2. Arylamide-Based Small Molecules

Based on the wide application of arylamides in the field of hydrogen bonding and molecular recognition and their relatively simple synthesis strategy, Degrado et al. designed and synthesized a series of arylamide foldamers to simulate the amphiphilic topological conformation of AMPs [42,43]. After vast quantities of structure optimization (such as replacement of the benzene ring with a pyrimidine ring, introduction of arginine residues, trifluoromethyl groups, tetrahydropyrrole groups, etc.), they successfully obtained a candidate compound **PMX-30063** (**4**, Figure 2) with good antibacterial activity for clinical research [44,45,46]. At present, **PMX-30063** has entered a phase III clinical trial for the treatment of acute bacterial skin infections caused by *S. aureus*.

### 2.3. Oligopeptide Derivatives

To date, many researchers have focused on the optimization or design of novel synthetic AMPs with desired properties. Unnatural amino acids (UAAs) are valuable building blocks and have been used to develop AMP mimetics with unique structural and physicochemical properties. The rational incorporation of UAAs has become a very promising approach to endow AMPs with strong and long-lasting activity but no toxicity. To break the limitation and expand the peptidomimetics family, Cai et al. developed a new class of peptidomimetics based on the backbone of peptide nucleic acids (PNAs) “*γ*-AApeptides” (Figure 3), which possess prominent advantages such as resistance to proteolytic degradation, enhanced chemodiversity, good selectivity, and outstanding bioactivity [47]. In addition, the secondary amines on the backbone can react with innumerable agents which are capable of balancing positively charged groups with hydrophobic residues. In the past decade, Cai’s group has conducted research on the design of small molecule antibacterial agents based on *γ*-AApeptides including hydantoins (**5**, Figure 3), acylated reduced amides (**6**, Figure 3), and biscyclic guanidines (**7**, Figure 3). *γ*-AApeptides as antimicrobial agents obtained excellent *in vitro* and *in vivo* bioactivity, good selectivity, and prevented the development of drug resistance in bacteria. The substitution of a urea moiety for the CH_2_-CO-NH units in the *γ*^4^-peptide (Figure 3) backbone by Gopi et al. is another series of peptoids [48,49,50].

Apart from the modification of amino acid backbones, the studies of Svendsen et al. showed that large substituent groups at the carbon terminus can effectively inhibit the hydrolysis of pancreatic enzymes and chymotrypsin by blocking the amide bond [51,52]. Based on this strategy, **LTX-109** (**8**, Figure 3) was designed and synthesized as a tripeptide derivative from bovine lactoferrin and is currently under clinical research [53,54,55]. The results of the phase I/II clinical study indicated that **LTX-109** is a promising novel antimicrobial agent with a low tendency to induce bacterial drug resistance during treatment of *S. aureus* infection during hospitalization [56].

### 2.4. Guanidine-Based Small Molecules

Guanidine derivatives have been exploited as privileged structural motifs in designing novel drugs for the treatment of various infectious diseases due to their hydrogen-bonding capability and protonatability at physiological pH in the context of interactions with biological targets [57]. As such, a brominated guanidinium oxazolidinone named synoxazolidinones A (**9**, Figure 4) exhibited antibacterial and antifungal activities, which was isolated from sub-Arctic ascidian Synoicum pulmonaria [58]. A research study also reported that guanidinium cations can promote the disruption of bacterial cytoplasmic membranes [59]. In 2015, Shaw et al. used a scaffold-ranking library to screen for antibacterial activity against the ESKAPE pathogens and identified a bis-cyclic guanidine library that displayed strong antibacterial activity [60]. The lead compound **10** (Figure 4) was broadly active against all ESKAPE organisms at concentrations <2 μM and had antibiofilm effects. In addition, **10** showed limited potential for the development of resistance and displayed low toxicity against human lung cells and erythrocytes. In 2017, Cai et al. developed a novel series of bis-cyclic guanidines as potent membrane-active antibacterial agents. Their results showed that lead compound **11** (Figure 4) killed Gram-positive bacteria with MIC values of 0.16–0.75 μg/mL and Gram-negative pathogens with MIC values of 1.5–6.0 μg/mL. As anticipated, this series of compounds was able to partially mimic the mechanism of AMPs by rapidly rupturing bacterial membranes [61]. Kumar et al. reported the synthesis of amphipathic guanidine-embedded glyoxamide-based compounds as antimicrobial agents via ring-opening reactions of *N*-naphthoylisatins with amines and amino acids [62,63]. These compounds were investigated for their antibacterial activity by the determination of minimum inhibitory concentration (MIC). The bromo-substituted guanidinium compound **12** (Figure 4) exhibited good MIC against *S. aureus* (3.9 μM) and *E. coli* (15.6 μM) and disrupted established biofilms of *S. aureus* by 83% at a concentration of 62.4 μM [64].

### 2.5. Xanthone-Based Small Molecules

Xanthones (**13**, Figure 5) are a family of *O*-heterocyclic symmetrical compounds with a dibenzo-*γ*-pyrone scaffold due to their structural variety and their biological activities and can be found in the marine environment, plants, fungi, and lichen [65,66,67]. In 2013, Beuerman et al. reported that α-mangostin (**14**, Figure 5), a natural xanthone extracted from a common southeast Asian fruit, Garcinia mangostana, disrupts the cytoplasmic membrane of Gram-positive organisms including MRSA [68]. Based on α-mangostin, they further developed a series of xanthone derivatives by cationic modification of the free hydroxyl groups to improve membrane selectivity [69]. The results indicated that compound **15** (Figure 5) exhibited potent antimicrobial properties against Gram-positive bacteria and killed bacteria rapidly without inducing drug resistance. Biophysical studies and molecular dynamics simulations revealed that **15** targets the bacterial inner membrane, forming an amphiphilic conformation at the hydrophobic–water interface.

### 2.6. Other Small Molecule Peptidomimetics

In addition to the structural scaffolds mentioned above, there are various other types of small molecule peptidomimetics, both natural and synthetic. In 2012, Xu et al. reported the isolation of ianthelliormisamines A–C from the marine sponge [70], and Brunel et al. designed ianthelliormisamine synthetic analogues as antibiotics based on this scaffold [71]. They performed modification on the positively charged groups and found compound **16** (Figure 6) was able to affect the susceptibility of bacteria to commercial antibiotics used in clinic. Further mechanism studies showed that compound **16** could cause bacterial cell membrane depolarization and inhibit the drug efflux pump. In 2016, Woster et al. described a series of synthetic diamines as broad-spectrum bactericidal agents which could reduce biofilm formation and promote biofilm dispersal in *P. aeruginosa* [72]. The most potent analogue, compound **17** (Figure 6), primarily acts by depolarization of the cytoplasmic membrane and permeabilization of the bacterial outer membrane which was confirmed by transmission electron microscopy. In 2014, Haldar et al. developed and reported phenylalanine-conjugated lipophilic norspermidine derivatives with excellent antibacterial activity [73]. Their structure–activity relationship study also interpreted how the incorporation of an aromatic amino acid drastically improved selective antibacterial activity. Compound **18** (Figure 6) exhibited a rapid bacterial killing rate and induced no resistance. In 2023, Lin et al. reported that phenothiazine derivatives biomimicking AMPs were designed and synthesized [74], which was inspired by the amphiphilic structure and function of AMPs and the good druggability of the phenothiazines. The most promising compound **19** (Figure 6) bearing an n-heptyl group and two arginine residues displayed potent bactericidal activity and showed low hemolysis activity (HC_50_ = 281.4 ± 1.6 μg/mL) and low cytotoxicity (CC_50_ > 50 μg/mL) toward mammalian cells, as well as potent *in vivo* efficacy in a murine model of bacterial keratitis. They also reported a series of indole-based amphiphilic antimicrobial peptidomimetics with hydrophobic side chains and hydrophilic cationic moieties back in 2021 [75]. Among these derivatives, compound **20** (Figure 6) demonstrated the most potent antimicrobial activity. In 2021, Bayer et al. designed and synthesized a series of amphipathic barbiturates as mimics of AMPs and the marine natural products Eusynstyelamides [76]. These barbiturate derivatives consist of an achiral barbiturate scaffold with two cationic groups and two lipophilic side chains. MICs of 2–8 μg/mL were achieved against 30 multi-resistant clinical isolates of both Gram-positive and Gram-negative bacteria. Moreover, the lead compound **21** (Figure 6) demonstrated potent *in vivo* efficacy in a neutropenic peritonitis mice model with clinical isolates of *E. coli* and *K. pneumoniae*.

## 3. AMP-Inspired Antimicrobial Cationic Oligomers and Polymers

Well-known AMPs are amphiphilic chains consisting of 12–50 amino acid residues and a variable charge of +1 to +10 from the cationic residues [77]. Positively charged groups allow for selective binding to bacterial cells and hydrophobic groups cause insertion into and disruption of the phospholipid cell membrane [78]. Based on the membrane-disruptive antimicrobial mechanisms, synthetic polymers mimicking AMPs have emerged as novel antimicrobial candidates [79]. These polymers have shown good broad-spectrum antimicrobial activity, rapid bactericidal kinetics, and a low propensity to induce resistance. In contrast to AMPs, cationic polymers are less susceptible to proteolysis and more beneficial for economic large-scale manufacturing through automated processes [80]. Systematic optimization of antimicrobial cationic polymers has been investigated by researchers in recent years, such as the polymer composition, chain length, hydrophobicity, and cationic charge [81,82].

### 3.1. β-Peptide Derivative Polymers

In 2019, Liu et al. designed and synthesized a novel series of AMPs mimicking polymers which are composed of unnatural β-amino acids [83]. The best poly-β-peptide (20:80 Bu:DM), compound **22** (Figure 7), displays potent and broad-spectrum antibacterial activity against antibiotic-resistant super bugs and low toxicity toward mammalian cells. Moreover, compound **22** could kill bacteria quickly within 5 min and the bacterial strains develop no resistance even over 1000 generations. Further *in vivo* studies indicated that **22** outperformed antibiotics in the removal and reduction of the viability of established biofilms, achieving a maximum activity of around 80% reduction in viability. Moreover, **22** also exhibited immunomodulation in inducing chemokines and anti-inflammatory cytokines and suppressing LPS-induced proinflammatory cytokines, as well as reducing tissue dermonecrosis [84]. Most positively charged moieties utilized primary amines which are inspired by lysine residues. Liu et al. then explored the impact of amine group variation (primary, secondary, or tertiary amine) on antibacterial performance. Their studies showed that a secondary ammonium was superior to either a primary ammonium or a tertiary ammonium in antibacterial β-peptide polymers [85]. The optimal polymer **23** (Figure 7) displays potent activity against antibiotic-resistant bacteria and high therapeutic efficacy in treating MRSA-induced wound infections and keratitis as well as low acute dermal toxicity and low corneal epithelial cytotoxicity. They also developed poly(DL-diaminopropionic acid) (PDAP) from the ring-opening polymerization of β-amino acid *N*-thiocarboxyanhydrides, by mimicking ε-poly-lysine [86]. Their studies showed that PDAP kills fungal cells by penetrating the fungal cytoplasm, generating reactive oxygen, and inducing fungal apoptosis. The optimal PDAP compound **24** (Figure 7) displays potent antifungal activity with an MIC value as low as 0.4 µg/mL against *Candida albicans*, and with negligible hemolysis and cytotoxicity.

Gellman et al. developed random poly-β-peptide copolymers via ring-opening polymerization of β-lactams and documented structure–activity relationships in this polymer family (compound **25**, Figure 7) [87,88]. Their research indicated that these heterogeneous polymers in terms of subunit order and stereochemistry displayed comparable biological activities to those that have been documented among host-defense peptides and analogous synthetic peptides. Tew et al. developed a novel synthetic approach for synthetic mimics of antimicrobial peptides through ring-opening metathesis polymerization (ROMP) and deprotection [89,90]. Accordingly, the created polymers **26** and **27** displayed biological activity similar to natural proteins, including antimicrobial and cell-penetrating peptides.

### 3.2. Poly(2-oxazoline)

In 2020, for the first time, Liu et al. demonstrated that poly(2-oxazoline) (POX) (Figure 8) can work as a functional mimic of peptides, which opens new avenues in designing and exploring peptide mimetics for biological functions and applications [91]. POX-based glycine pseudopeptides showed potent activities against various pathogens. They further design and synthesize guanidinium-functionalized poly(2-oxazoline)s by mimicking cell-penetrating peptides and obtained polymer **28** (Figure 8) [92]. Polymer **28**, bearing a methylene spacer arm, displays potent activities against the drug-resistant fungi and biofilm, negligible toxicity, and insusceptibility to antimicrobial resistance. Most importantly, polymer **28** can break blood–brain barrier (BBB) retractions to exert promising antifungal functions in the brain and demonstrates potent *in vivo* antifungal therapeutic efficacy in mouse models including skin infections, systemic infections, and meningitis.

### 3.3. Polycarbonates

In 2014, Yang et al. reported cationic polycarbonates containing quaternary ammonium salts as antimicrobial agents [93]. This series of polycarbonates containing propyl and hexyl side chains quaternized with various nitrogen-containing heterocycles, such as imidazoles and pyridines (**29**, Figure 9). These polymers demonstrate a wide spectrum of activity against Gram-positive, Gram-negative, and fungal pathogens. Later in 2017, Cai et al. reported the design and synthesis of amphiphilic polycarbonates containing primary amino groups [94]. Polymer **30** (Figure 9) exhibited potent antimicrobial activity and excellent selectivity to Gram-positive bacteria, including multidrug-resistant pathogens. Fluorescence and TEM studies suggest that these polymers are likely to kill bacteria by disrupting bacterial membranes and show a low tendency to elicit resistance in bacteria. They further developed lipidated antimicrobial guanidinylate polycarbonates by connecting a saturated lipid chain in 2024 [95]. These polymers showed potent antimicrobial activity against a panel of bacteria with fast-killing kinetics and low resistance-development tendencies. The optimal polymer **31** (Figure 9) showed excellent antibacterial activity against *C. difficile* infection (CDI) *in vivo* via oral administration. In addition, compared with vancomycin, the polymer demonstrated a much-prolonged therapeutic effect and virtually diminished recurrence rate of CDI.

### 3.4. Polysulfoniums

In 2021, Rao et al. developed a new type of antibacterial polysulfoniums with cationic sulfoniums and alkane spacers which were all installed in the polymer main chain [96,97]. These polysulfoniums presented effective activity against planktonic fungi and bacteria with MICs of 0.5–32 μg/mL. The polysulfonium homopolymer **32** (Figure 10) can provide an 80–90% reduction in biofilm mass and >99% killing of *C. albicans* and *E. coli* cells in 3-day mature biofilms at 2 × MIC as well as steadily low hemolytic toxicity. Polysulfonium **33** (Figure 10) exhibited excellent antibacterial activity against a broad spectrum of clinically relevant bacteria with MICs in the range of 1.25–10 μg/mL, as well as negligible hemolytic effects at polymer concentrations even up to 10,000 μg/mL. In 2020, Sun et al. reported a convenient approach to synthesize a novel type of polypeptoid containing both sulfonium and oligo(ethylene glycol) (OEG) moieties by ring-opening polymerization and a post-modification strategy [98]. The obtained polypeptoid sulfonium salts exhibited excellent antibacterial activity against *S. aureus* with MICs in the range of 3.9–7.8 μg/mL. Remarkably, the as-prepared polysulfonium **34** (Figure 10) showed rapid and potent antibacterial activity within 5 min. They further performed a systematic investigation and prepared polysulfonium **35** (Figure 10) in order to explore the influence of the overall hydrophobic/hydrophilic balance on the antimicrobial activity and selectivity [99]. The obtained polypeptoid sulfoniums, with high selectivity and potent antibacterial properties, are excellent candidates for antibacterial treatment and open up new possibilities for the preparation of a class of innovative antimicrobials.

### 3.5. Polyisocyanate Copolymers

Polyisocyanate (Figure 11) composed of a 1-nylon backbone can adopt a helical conformation, which is well-suited to mimic the structures and functions of natural peptides [100]. By introducing a bulky aromatic group into monomers, steric and electron-withdrawing effects can be increased, facilitating a tendency of cross-propagation between the monomers. Therefore, isocyanates with large aromatic pendant groups should be investigated to obtain alternating polyisocyanate copolymers [101]. Accordingly, Lee et al. developed polyisocyanate copolymer **36** (Figure 11) in 2022. Their studies showed that a concentration of 313–1250 μg/mL was required to reduce the colony forming units (CFUs) of bacteria to <100 CFU/mL and the cationic amphiphilic structures can facilitate interactions with the bacterial membrane. The alternating polyisocyanate copolymer with an amphiphilic helical conformation paves the way to mimic the structures and functions of natural α-helical peptides.

## 4. Mechanism of Action

Since the 1980s, AMPs have been known to disrupt the integrity of bacterial membranes [102]. The putative mechanism of action is often described in molecular detail by the Shai–Matsuzaki–Huang (SMH) model [103,104]. The bacterial membrane is rich in negatively charged acidic phospholipids which can bind to cationic antimicrobial peptides through electrostatic adsorption. The hydrophobic region of antibacterial peptides can aggregate with the surface of amphoteric phospholipids of the cell membrane and accumulate on the membrane surface. Three models have been proposed accordingly: the carpet model, toroidal pore model, and barrel-stave model (Figure 2) [105,106]. In recent years, with the development of biophysics and molecular biology, the mechanism of action of AMPs has been further verified.

Model membranes were widely utilized by researchers to probe the interaction of AMPs with phospholipid bilayers. A classical experiment to evaluate membrane permeabilization induced by AMPs is to monitor the release of a fluorescent dye entrapped within a liposome upon mixing with a diluted AMP solution [107,108]. Model membranes include artificial bilayer lipid membranes (BLMs), large unilamellar vesicles (LUVs), large unilamellar vesicles (LUVs), and giant unilamellar vesicles (GUVs).

Molecular dynamics (MD) simulation is another widely used approach and is rapidly emerging as a very powerful tool in investigating precise detail at the molecular level [109]. In 2006, Klein et al. reported that the formation of strong interactions between the lipid headgroups and the amine groups of the cationic polymers assists in the initial association with the lipids. However, the primary driving force for the observed partial insertion appears to be the hydrophobic effect [110]. Furthermore, in 2022, Lee et al. identified the bacterial membrane selectivity of AMPs via MD and the results showed that polar residues in AMPs bound to a model mimicking the bacterial inner/outer membranes preferentially over the eukaryotic plasma membrane [111].

Direct observation of the disruption of bacterial cell membranes provides the most convincing evidence for mechanism of action, and such visualization has been conducted extensively by scanning electron microscopy (SEM) and transmission electron microscopy (TEM) [112,113,114]. In 2020, high-speed atomic force microscopy (HS-AFM) enabled the first-ever visualization of the activity of antimicrobial peptides on membranes at the molecular level and confirmed the hypothesis model so far: daptomycin oligomerization and the formation of a half pore [115]. In 2023, Chen et al. reported that cryo-electron tomography (cryo-ET) directly visualized how the antimicrobial peptide damages *E. coli* cell membranes via a carpet/detergent-like mechanism (Figure 3) [116]. In the same year, the mechanism of action of the antimicrobial cationic tripeptide **LTX-109** (**8**) was verified by a combination of HS-AFM and MD simulations [117].

## 5. Conclusions and Outlook

The antimicrobial peptide database has catalogued more than 2600 natural antimicrobial peptides [118]. Among them, polymyxin B (**37**) and daptomycin (**38**) (Figure 12) are well known marketed drugs as the two “last-resort” antibiotics [119,120]. It is especially noteworthy that polymyxins are increasingly being used as last-line therapy to treat otherwise untreatable serious infections caused by Gram-negative bacteria that are resistant to essentially all other currently available antibiotics.

The development of antimicrobial drugs with novel structures and clear mechanisms of action that are active against drug-resistant bacteria has become an urgent need in safeguarding human health due to the rise of bacterial drug resistance. The fascinating discovery of AMPs and the development of amphipathic peptidomimetics have lay the foundation for novel antimicrobial agents to combat drug resistance. Researchers have undertaken great endeavors to overcome the drawbacks of AMPs, such as the poor stability and high production cost. Antibacterial small molecule peptidomimetics and peptide-mimic cationic oligomers/polymers have become the major approaches and there have been dozens of successful compounds entering clinical trials (Table 1) [121,122].

Structure–activity relationship studies of amphipathic peptidomimetics mainly focused on the balance of charge and hydrophobicity [123,124]. The net positive charge contributes to the selectivity due to the negatively charged bacterial membranes. Increased hydrophobicity will improve antimicrobial activities to a certain extent, but tends to cause stronger hemolytic activity in the meantime. As the field of peptide-mimetic antimicrobial small molecules and oligomers/polymers has continuously expanded, mechanism-of-action studies also play a critical role for guiding further rational design of molecular structures. We look forward to continued growth and future developments in this exciting interdisciplinary field; however, potential pitfalls in the development of novel antimicrobial agents are certainly no exception: pharmacokinetics, long-term toxicity, and biodistribution are all challenges that will likely require sustained efforts to surmount.

## Data Availability

Not applicable.

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
