# Peer review of "Recent Advances in Amphipathic Peptidomimetics as Antimicrobial Agents to Combat Drug Resistance"

_molecules, 2024, doi:10.3390/molecules29112492_

Round 1

Reviewer 1 Report

Comments and Suggestions for Authors

In this review article, the authors revisit the recent progress on amphipathic peptidomimetics as anti-microbial agents. Peptidomimetics are emerging as new modalities for drug discovery due to its stability and specificity. Amphiphilic peptides can have balance between hydrophobicity and hydrophilicity, and thus it makes ideal candidate for antimicrobial agent. This review article well written and some points need to be discussed before its formal acceptance:

Points:

1.    The authors should emphasis more on the amphiphilicity of peptides and why this is essential for the design of peptide based antimicrobial agent.

2.    Author should also discuss the potential application of hybrid peptide for the design of antimicrobial agents.

3.    The mode of action section needs to more elaborated and should add some schematic figure to better understand to the reader for how amphiphilicity of peptides needed.

4.    Some of the molecular structure need to be enlarged for proper visualizations.

5.    Author should also include other peptidomimetics structure such as urea based for antimicrobial agent. They should also consider other secondary structure such as beat hairpin.

6.    Author should compare the peptidomimetics structure with naturally occurring AMPs.

Author Response

Reviewer 1:

In this review article, the authors revisit the recent progress on amphipathic peptidomimetics as anti-microbial agents. Peptidomimetics are emerging as new modalities for drug discovery due to its stability and specificity. Amphiphilic peptides can have balance between hydrophobicity and hydrophilicity, and thus it makes ideal candidate for antimicrobial agent. This review article well written and some points need to be discussed before its formal acceptance:

We thank the reviewers for their comments.

  1. The authors should emphasis more on the amphiphilicity of peptides and why this is essential for the design of peptide based antimicrobial agent.

The importance of amphiphilicity has been added to the manuscript as suggested. Related publications are also included.

  1. Author should also discuss the potential application of hybrid peptide for the design of antimicrobial agents.

This review is mainly focused on AMP inspired antimicrobial peptidomimetics instead of peptides. Hybrid peptide are based on peptoids and peptides, and we have included antimicrobial peptoids (section 2.3) in this review.

  1. The mode of action section needs to more elaborated and should add some schematic figure to better understand to the reader for how amphiphilicity of peptides needed.

Schematic figure for mode of action has been added. Please see Figure 1 and Figure 2.

  1. Some of the molecular structure need to be enlarged for proper visualizations.

We thank for the reviewer’s suggestion. We have enlarged some of the schemes for better visualizations.

  1. Author should also include other peptidomimetics structure such as urea based for antimicrobial agent. They should also consider other secondary structure such as beat hairpin.

Urea-based peptidomimetics structure has included in the manuscript. Please see Scheme 3.

Beta-hairpin peptides are also an important class of AMPs with unique beta-turn secondary structure. The main focus of this manuscript is the backbone design of peptoids and polymers, and 2nd structure of protein will bring out a whole research field.

  1. Author should compare the peptidomimetics structure with naturally occurring AMPs.

Structure of natural AMPs have been added to the manuscript. Please see in Scheme 12.

Reviewer 2 Report

Comments and Suggestions for Authors

Dear Authors,

The document entitled "Recent Advances on Amphipathic Peptidomimetics as Antimicrobial Agents to Combat Drug-resistance" focuses on the development of antimicrobial drugs with novel structures and clear mechanisms of action against drug-resistant bacteria. Amphipathic peptidomimetics, derived from natural antimicrobial peptides (AMPs), offer effective general antimicrobial activities and a unique membrane-action mechanism. It emphasizes the need to overcome the limitations of AMPs through the research and development of new peptidomimetic antimicrobial agents. It is clearly described that the global increase in antibiotic resistance represents a significant challenge, highlighting the urgent need to develop new antibacterial molecules that can overcome inherent, acquired, and adaptive resistance mechanisms. Small molecule peptidomimetics, which mimic the action of natural antimicrobial peptides (AMPs), emerge as promising candidates, noted for their lower molecular weight, enhanced stability, and simplified synthesis, with several already in clinical trials such as CSA-13, PMX30063, and LTX109. Concurrently, AMP-inspired cationic polymers and oligomers offer additional advantages such as reduced susceptibility to degradation and the ability to be produced on a large scale, showing effective antibacterial activity and a low propensity to induce resistance. The mechanism of action of these compounds, explained through the Shai-Matsuzaki-Huang model, involves the destabilization of bacterial membranes through pore formation. This detailed understanding of the underlying mechanism is essential for the rational design of future antimicrobial agents, underscoring the importance of continuing the development and evaluation of new peptidomimetics and polymers in the fight against antibiotic resistance.

The manuscript requires some minor improvements for publication, which include correcting italicized words for the abbreviation et al. when referring to an author, the word in vivo when detailing a type of assay, and italicization of letters in chemical names; here are just a few examples from the lines, but I encourage the authors to review the document in its entirety.

line 294, italicize et al. line 261, italicize the letter N in N-thiocarboxyanhydrides line 246, italicize in vivo

The molecules presented throughout the document to illustrate structural complexity do not provide relevant information within the document, so it is suggested to create a table that groups the most relevant structures, indicating the family, most notable properties highlighted, and the current state of research. This would help provide a global view of the current state of the area.

After making these improvements, I recommend that the work be publishable.

Author Response

Reviewer 2:

The manuscript requires some minor improvements for publication, which include correcting italicized words for the abbreviation et al. when referring to an author, the word in vivo when detailing a type of assay, and italicization of letters in chemical names; here are just a few examples from the lines, but I encourage the authors to review the document in its entirety.

We thank the reviewers for their comments.

  1. line 294, italicize et al. line 261, italicize the letter N in N-thiocarboxyanhydrides line 246, italicize in vivo

We thank the review for pointing out the formatting issues. The whole context has been reviewed and corrected. Please see highlighting sections in the revised manuscript.

  1. The molecules presented throughout the document to illustrate structural complexity do not provide relevant information within the document, so it is suggested to create a table that groups the most relevant structures, indicating the family, most notable properties highlighted, and the current state of research. This would help provide a global view of the current state of the area.

We thank the reviewer for the suggestion. The summary table 1 has been added to the manuscript accordingly.

After making these improvements, I recommend that the work be publishable.